# Respiratory Vaccines in Older Adults: A Bibliometric Analysis and Future Research Agenda

**DOI:** 10.3390/vaccines13030240

**Published:** 2025-02-26

**Authors:** Jose F. Parodi, Fernando Runzer-Colmenares, Carlos Cano-Gutiérrez, José Luis Dinamarca-Montecinos, Patricio Bendía-Gomez de La Torre, Paulo Fortes Villas Boas, Javier Flores-Cohaila, Diego Urrunaga-Pastor, Luis Miguel Gutiérrez-Robledo

**Affiliations:** 1Academia Latinoamericana de Medicina del Adulto Mayor—ALMA, 28001 Madrid, Spain; jparodig@usmp.pe (J.F.P.); frunzer@cientifica.edu.pe (F.R.-C.); carlosalbertocano@gmail.com (C.C.-G.); josedinamarcam@gmail.com (J.L.D.-M.); patriciogeriatra@hotmail.com (P.B.-G.d.L.T.); paulo.boas@unesp.br (P.F.V.B.); gutierrezrobledoluismiguel@gmail.com (L.M.G.-R.); 2Facultad de Medicina, Centro de Investigación del Envejecimiento, Universidad de San Martín de Porres, Lima 15011, Peru; 3CHANGE Research Working Group, Carrera de Medicina Humana, Universidad Científica del Sur, Lima 15067, Peru; durrunaga@cientifica.edu.pe; 4Instituto de Envejecimiento, Facultad de Medicina, Pontifica Universidad de Javeriana, Bogotá 110231, Colombia; 5Hospital Dr. Gustavo Fricke, Universidad de Valparaíso, Viña del Mar 2362804, Chile; 6Facultad de Medicina, Pontificia Universidad Católica del Ecuador, Quito 170143, Ecuador; 7Medical School of Botucatu, São Paulo State University, São Paulo 18618-687, Brazil; 8Grupo NEMECS: Neurociencias, Metabolismo, Efectividad Clínica y Sanitaria, Carrera de Medicina Humana, Universidad Científica del Sur, Lima 15011, Peru; 9Instituto Nacional de Geriatría, 10200 Ciudad de México, Mexico

**Keywords:** respiratory vaccines, older adults, Latin America, bibliometric analysis, influenza, pneumococcal, COVID-19

## Abstract

Background/Objectives: Respiratory infections impact older adults due to immunosenescence and comorbidities, resulting in increased healthcare costs and mortality. While vaccination is a critical preventive measure, research on respiratory vaccines in older adults in Latin America and the Caribbean (LAC) remains underexplored. This study aims to map the research landscape and identify emerging themes to guide future studies. Methods: A bibliometric analysis was conducted using the Web of Science database, focusing on publications up to 2023 related to respiratory vaccines in LAC’s older adult population. PRISMA-ScR guidelines were followed for data extraction and analysis, with performance metrics and scientometric mapping conducted using Biblioshiny 4.1 and VOSviewer. Results: Ninety-nine studies spanning forty-one journals and 575 authors were included. Brazil contributed 70% of publications, followed by Mexico and Argentina. Influenza and pneumococcal vaccines were the most studied, focusing on coverage, acceptance, and cost-effectiveness. Emerging themes included COVID-19 vaccine effectiveness and vaccination-associated factors. Brazil was identified as the primary hub for collaboration across the region, while other countries made limited contributions. Conclusions: The findings highlight disparities in research output, with Brazil dominating and significant gaps in other LAC countries. Future research should prioritize genomic studies, vaccine efficacy in comorbid populations, and adaptive immunization strategies. Building research capacity and fostering international collaborations are essential for improving vaccination outcomes in older adults across LAC.

## 1. Introduction

Respiratory infections significantly impact older adults’ health globally [1]. Age-related changes and comorbidities increase morbidity, mortality, and healthcare costs. Immunosenescence, chronic diseases, and frailty increase the susceptibility of older adults to severe respiratory infections and complications [2,3]. Therefore, healthcare systems must prioritize prevention, early detection, and specialized management of respiratory diseases in this vulnerable population.

Prevention and proper control of respiratory infections in older adults are essential to reduce their impact [4]. Preventive measures include influenza and pneumococcal vaccination [5,6,7], promoting a healthy lifestyle, and controlling risk factors like smoking [8]. Furthermore, basic infection prevention practices, such as hand washing, mask wearing, and physical distancing, are crucial in long-term care settings and nursing homes, where respiratory infections can spread quickly [9].

Among several strategies, vaccinations have emerged as the key solution. Although these have been quickly adopted worldwide, the Latin America and the Caribbean (LAC) region has faced several shortcomings [10]. These range from disparities in health insurance systems [11,12] to a lack of a properly trained health workforce [13]. Based on what was described above, vaccination research in this group has remained a neglected topic. The increasing burden of respiratory infections in older adults, exacerbated by immunosenescence and the presence of comorbidities, underscores the urgent need to investigate effective vaccination strategies in this vulnerable population. Despite the critical importance of vaccines, studies on their implementation and effectiveness in older adults in Latin America and the Caribbean is scarce. This study not only provides a bibliometric analysis revealing current trends and research gaps but also establishes an agenda for future research that could significantly improve health outcomes in this population. Furthermore, the lack of systematization in the current body of literature hinders our understanding of this forgotten field. Therefore, to fill this gap, we decided to conduct this study. Here, our objective was to map and delimit the field of respiratory vaccines for older adults in the LAC region through a bibliometric analysis.

## 2. Materials and Methods

This bibliometric analysis was conducted following the four-step approach proposed by Öztürk (definition of the research aim, data collection, data analysis, and interpretation of findings [14]) and was reported adhering to the PRISMA Extension for Scoping Reviews statement to enhance reproducibility [15].

The PCC (Population, Concept, and Context) framework was used to develop research questions. Hence, the primary research question was “What is the state of the research field of respiratory vaccines for geriatrics in Latin America and the Caribbean?” The secondary research questions were the following: (RQ1) Which journals, countries, and publications are the most impactful in the field? (RQ2) What are the networks of collaborations between countries? (RQ3) What are the most researched topics based on the authors’ keywords? (RQ4) What is the conceptual structure of the field? (RQ5) What are emerging themes?

The researchers selected the Web of Science (WoS) database for their bibliometric analysis. WoS is widely recognized as the most extensive global citation database and the most used for this type of research [16]. Although some scholars have favored Scopus, earlier comparisons have indicated that both databases have comparable journal coverage [17,18].

The search strategy was as follows: ((TS = (Geriatric* OR gerontology OR “older people” OR “elderly” OR “seniors”) OR TI = (Geriatric* OR gerontology OR “older people” OR “elderly” OR “seniors”) OR AK = (Geriatric* OR gerontology OR “older people” OR “elderly” OR “seniors”)) AND ((TS = (“Vaccin*” OR “Inmunogen*”)) AND (TS = (Respirator* OR Corona* OR Influenza OR Pneumo*))) AND CU = (Argentina OR bolivia OR brazil OR brasil OR chile OR colombia OR “Costa Rica” OR Cuba OR ecuador OR “El Salvador” OR guatemala OR haiti OR Honduras OR mexico OR Mejico OR nicaragua OR panama OR paraguay OR peru OR “Puerto Rico” OR “Dominican Republic” OR uruguay OR Venezuela OR “Latin America” OR Caribbean). These terms were only searched on the Science Citation Index Expanded, Social Sciences Citation Index, and Emerging Sources Citation Index. Filters were (1) studies published until 2023; (2) articles and reviews; and (3) studies in English, Spanish or Portuguese. The last search was performed on 1 May 2024.

Retrieved studies were downloaded and screened to enhance specificity. One author (JF) reviewed each study title and abstract to assess if they met the eligibility criteria. The criteria were (1) studies focused on respiratory vaccines and (2) studies with older adults as their population. Retracted studies or conference proceedings were excluded. In case of discrepancies, they were solved through discussion until a consensus was reached. With the list of eligible studies, these were re-searched in the WoS database, employing their unique IDs. The eligible studies were then downloaded in a text file with their complete information and cited references for analysis.

Data analysis was conducted using performance analysis and scientometric mapping [19]. The performance analysis was conducted using the Biblioshiny 4.1 app (https://www.bibliometrix.org/home/index.php/layout/biblioshiny, accessed on 28 April 2024) from the bibliometrix package in Rstudio (Boston, MA, USA, 2024.12.1 version) [20].

We described the most impactful countries, journals, and studies to answer RQ1. For this, we defined impact as evidence of contributions to advance the field [21]. Hence, the following metrics, in descending order, were considered to assess impact: (1) H-index: a measure that assesses productivity and citations, it is defined as the number of published papers (*n*) by the researcher that have each been cited at least n times by other papers and is calculated as this value; (2) Total citations: the number of citations that a study has received both from this dataset (total local citations) and among all studies in the selected database (total global citations) [22].

To answer RQ2, a co-authorship co-occurrence analysis based on countries was performed in VosViewer [23]. Due to each country’s low number of publications, the network was constructed of countries with at least one publication. Furthermore, countries outside the LAC region were removed from this analysis, and the fractal counting approach was used, as it is superior to complete counting methods [24,25].

To answer RQ3, a keyword frequency analysis was performed. For RQ4 and RQ5, the VosViewer software was used (1.6.20 version, https://www.vosviewer.com/, accessed on 28 April 2024) [23]. In the case of RQ4, a keyword co-occurrence analysis was performed using a fractional counting approach with keywords that had over two occurrences. To interpret this graph, previous authors have suggested that each cluster can be named using a combination of their representative keywords logically [26]. Hence, that approach was used.

Meanwhile, for RQ5, a bibliographic coupling analysis was performed. It measures overlap in the citations of the publications themselves. Therefore, if two publications have the same (or similar) set of references, these are coupled and cover similar topics [27]. Furthermore, this analysis represents the present research field [28]. This was performed using a fractional counting approach on documents with over ten citations.

This research did not involve humans and used publicly available data, so ethical board approval was deemed unnecessary.

## 3. Results

After screening, 99 studies were eligible for the bibliometric analysis. These were published across 41 journals and represented 575 authors. The timeframe was 2004 to 2023, with 2009 being the year with the fewest publications (*n* = 1, 1%) and 2013 with the most (*n* = 12, 12%). The annual growth rate was 4.94%.

### 3.1. Performance Analysis

#### 3.1.1. The Journals

Table 1 shows the most impactful journals. These top 10 journals cover over half of the publications (*n* = 58, 58.5%). The scope of these journals is mainly public health (*n* = 7, 70%), followed by vaccines and infectious diseases (*n* = 3, 30%). When examining the country of origin from these top-ranked journals, most were from Brazil (*n* = 5, 50%). While the most impactful journal was *Vaccine* (H-index: 7, TC: 260, NP:10), the most productive journal was *Cuadernos de Saude Publica* (H-index: 5, TC:128, NP:13). Hence, the top-ranked journals are from Brazil, their scope is public health and they cover over half of publications.

#### 3.1.2. The Countries

Table 2 shows that 16 of the 33 countries from the LAC region were represented in the field and lists the top 10 countries with their key bibliometric indicators. The average number of documents per country was 7.5, with Brazil taking the lead with 70 papers, followed by Mexico (*n* = 14) and Argentina (*n* = 11). However, when adjusting per documents by corresponding authors, Brazil had sixty-three, Mexico nine, and the other countries less than three. It is worth noting that only five out of sixteen LAC countries have published more than two documents. Furthermore, only seven out of sixteen LAC countries had at least one document with a corresponding author.

The network of collaborations between countries from the LAC region is shown in Figure 1. Three clusters were identified. Cluster 1 (red) is the most prominent, with countries like Brazil, Mexico, Argentina, Chile, and Colombia, which have been the most impactful in previous analyses. Cluster 2 (green) comprises countries such as Cuba, Uruguay, Costa Rica, Panama, Peru, and El Salvador. These countries are characterized by low impact and low numbers of publications. Cluster 3 (blue) comprises Guatemala and Honduras. Furthermore, the most influential country was Brazil (Links: 15, Total link strength (TLS): 5); hence, it is the country that connects all clusters. Therefore, vaccine research in LAC is characterized by two major working groups: (1) the highly productive group led by Brazil, Mexico, Chile, Argentina, and Colombia, and (2) the bottom-end group led by other countries in the region.

#### 3.1.3. The Ideas

Table 3 shows the most impactful studies categorized by global and total citations. This classification represents the most important studies for a global audience (global citations) and the LAC audience (local citations). At first glance, no studies between groups were the same. Furthermore, the studies with the highest global citations were published between 2013 and 2021, while studies with the highest local citations were published between 2005 and 2011. Most studies in the worldwide group were conducted internationally, except one in Brazil. However, all studies in the local group were conducted in Brazil. Most studies in both groups centered on the influenza vaccine. While studies in the global group centered mainly on the effectiveness of vaccines, the studies in the local group were on vaccine coverage and attitudes toward vaccinations. Hence, the most prominent ideas in the top-ranked studies are the influenza and COVID-19 vaccines, their effectiveness, and coverage and attitudes toward vaccinations.

In addition, we conducted a frequency analysis of author keywords to provide further evidence for the previously identified ideas. Among 210 keywords, the most frequent based on the number of appearances were influenza, aged, influenza vaccines, elderly, vaccination, the health of the elderly, COVID-19, immunization, influenza vaccine, and immunization programs. Hence, the major topics in this analysis were influenza, the elderly, COVID-19, and immunization programs.

### 3.2. Scientometric Mapping

#### 3.2.1. Conceptual Structure

Figure 2 shows an author’s keyword co-occurrence analysis illustrating the field’s conceptual structure. The network was constructed using a fractal counting approach, including keywords with two or more occurrences. The keywords with the highest total link strength were aged and influenza. Eight clusters were identified. Cluster 1 (red), the most prominent, was about “coverage of influenza in the elderly”. Cluster 2 (green) was about “influenza and pneumococcal vaccine in Mexico”. Cluster 3 (blue) was about “Influenza vaccine in Brazil”. Cluster 4 (yellow) covered topics on “surveys of attitudes toward vaccination on the elderly”. Cluster 5 (purple) studies were on the “COVID-19 vaccine”. Cluster 6 (light blue) was on the “effect of vaccines in mortality and cardiovascular disease”. Clusters 7 (orange) and 8 (brown) were added to Clusters 1 and 5, respectively.

#### 3.2.2. Emerging Topics

Figure 3 shows the bibliographic coupling analysis of forty documents with over ten citations. Six clusters were found. Cluster 1 (red) is marked by topics such as implementing and evaluating the vaccine program. One study from this cluster was a systematic review to assess the effect of social determinants of health in older adults with influenza vaccine access [29]; the rest of the studies were conducted in Brazil and evaluated the prevalence of vaccination and its associated factors. For example, several authors have explored factors associated with vaccination in the elderly in Brazil. These studies found socioeconomic level, older age (>70), marital status, community participation, education and contact with healthcare workers, and remaining physically active [34,35,39,40,41,42] as the most associated factors with vaccination in this population. Similarly, Cluster 4 (yellow) covered the cost-effectiveness of influenza vaccination. For example, one study evaluated the persistence of the antibody response in the elderly population, categorized based on physical activity [43], while others assessed the effectiveness of influenza vaccines [32,44]. Hence, these clusters show emerging topics for the LAC region, specifically Brazil, the study of influenza vaccine coverage, its associated factors, and its cost-effectiveness.

Cluster 2 (green) comprises studies focusing on the pneumococcal vaccine. Studies inside this cluster covered aspects such as the persistence of the antibody response [45,46] and cost-effectiveness analyses of the vaccine [47,48,49,50]. Cluster 3 (blue) is the newest and represents COVID-19-related studies. These studies focused on the effectiveness of COVID-19 vaccines in Brazil [30,51]. Lastly, Cluster 5 (purple) represents studies on vaccine coverage and its associated factors in Mexico [52].

## 4. Discussion

### 4.1. Main Findings

We conducted a bibliometric study to explore the field of respiratory vaccines in LAC. Our main findings were as follows: (1) Brazil emerges as the lead in the field, followed by Mexico, Colombia, Argentina, and Chile, while the rest of the LAC countries published less than two studies; (2) Public health in terms of vaccine coverage and acceptance, and its cost-effectiveness are the most studied topics in the LAC region; and (3) Emerging topics are the influenza vaccine, pneumococcal vaccine, and the COVID-19 vaccine, as well as factors associated with vaccination among this population.

Although we followed multiple methods to corroborate our findings and adhered to reporting guidelines, our study may still have some limitations. While the decision to use the WoS database was heavily supported by the available literature, there is still a chance to have missed some relevant studies. Furthermore, as no research priorities were established for vaccines in the elderly, a deductive approach to interpreting our findings was not feasible. Hence, this adds particular subjectivity to our interpretations. Lastly, our findings reflect the state of research on respiratory vaccines in the elderly in LAC. Therefore, these may not be applicable to specific countries, as most of the research output comes from Brazil.

### 4.2. Implications of Findings

To our knowledge, this is the first study to map and delimit the field of respiratory vaccines for geriatrics in LAC countries. Hence, it has proximal and distal implications. Among the proximal, our findings highlight the need for further research in countries outside of Brazil. This could serve as a starting point for researchers. For the LAC region, we describe the topics of interest, such as influenza and the pneumococcal vaccine. Moreover, adherence and acceptance of these vaccines appear as regional research priorities. While these were the most critical topics, others were absent, such as genomics, impact on mortality, and other emerging respiratory vaccines.

Several directions must be taken to address this lack of research. First, capacity building programs, funding opportunities, and international collaborations are needed to foster a balanced regional research landscape. Second, there is much to learn from countries such as Brazil and Mexico, which are leaders in the research field. The valuable experience of these researchers could aid in leading international research collaborations and academic networks. Third, a significant priority for decision-makers, policymakers, and academia should be the development of a shared research agenda on respiratory vaccines for the elderly. As detailed above, current research remains fragmented and may not cover critical areas for the region, such as specific sociodemographic conditions, frailty, or multimorbidity. Future research in Latin America should focus on addressing essential gaps in vaccination strategies for older adults. Studies exploring person-centered vaccination programs that account for comorbidities are crucial, as they can improve vaccine efficacy and safety in this population. In addition, investigating the impact of emerging diseases such as COVID-19, monkeypox, and dengue on vaccination protocols for the elderly is crucial for developing adaptive immunization strategies. Research on practical education and awareness campaigns tailored to older adults could improve vaccine acceptance and coverage. Comparative studies between urban and rural vaccination strategies may reveal disparities and inform targeted interventions. Exploring innovative technologies to improve vaccine coverage and monitoring in older adults could lead to more efficient immunization programs. Lastly, investigating the relationship between frailty, immune response, and vaccine effectiveness in older adults is vital to optimizing vaccination outcomes in this vulnerable population. These research directions will contribute to a more comprehensive and practical approach to vaccination in older adults across Latin America, ultimately improving health outcomes and quality of life for this growing demographic group.

This study contributes significantly to the literature by providing a comprehensive bibliometric analysis of respiratory vaccine research in older adults across Latin America and the Caribbean, highlighting critical gaps and disparities in research output. To strengthen its theoretical and empirical support, future research should incorporate frameworks on immunosenescence, recent epidemiological data on respiratory disease burdens, and comparative studies with other regions. Additionally, the implications for public health policies, the importance of international collaborations, and strategies for enhancing research capacity are essential for addressing the identified gaps and improving vaccination outcomes in this vulnerable population.

Furthermore, healthcare managers and policymakers can take several actionable steps based on this study’s findings. These include establishing dedicated funds for respiratory vaccine research in older adults, prioritizing countries with lower scientific output, and creating international collaboration programs. Implementing targeted awareness campaigns and developing culturally adapted vaccination strategies can improve vaccine coverage. Policies should be updated based on recent evidence and integrated into existing primary care programs. Investing in geriatric and vaccinology training for healthcare professionals and enhancing epidemiological surveillance infrastructure are crucial. These actions can help translate research findings into tangible improvements in older adults’ health across Latin America and the Caribbean, addressing the identified gaps and disparities in vaccine research and implementation [53]. Barriers like attitudes towards vaccines in Latin America are diverse. While many recognize their efficacy in preventing diseases, barriers such as lack of information, structural issues, and religious beliefs affect vaccination intentions. Trust in the healthcare system and government, as well as public accessibility, are key factors that increase willingness to vaccinate. However, conspiracy theories and misinformation can significantly decrease vaccination intention in the region [54].

On the other hand, due to the heterogeneous manner in which countries report their vaccination coverage by life stages, it is difficult to make a comparison of vaccination rates across regions. However, the present study highlights that in Latin America and the Caribbean research with funding is scarce, and it is necessary to take action to strengthen public health initiatives in this regard.

## 5. Conclusions

Several key findings emerge from this bibliometric analysis of respiratory vaccines for older adults in Latin America and the Caribbean (LAC). Brazil dominates the research landscape, followed by Mexico, Colombia, Argentina, and Chile, while other LAC countries make minimal contributions. The research focuses primarily on public health, particularly vaccine coverage, acceptance, and cost-effectiveness. Emerging topics include studies on influenza, pneumococcal and COVID-19 vaccines, and factors associated with vaccination among the elderly. This analysis highlights significant research gaps, particularly in countries outside Brazil, and underscores the need for increased research capacity, funding opportunities, and international collaborations across the LAC region. Furthermore, it emphasizes the importance of developing a shared research agenda on respiratory vaccines for the elderly. It addresses critical areas such as genomics, mortality impact, and emerging respiratory vaccines currently underrepresented in the literature.

## Figures and Tables

**Figure 1 vaccines-13-00240-f001:**
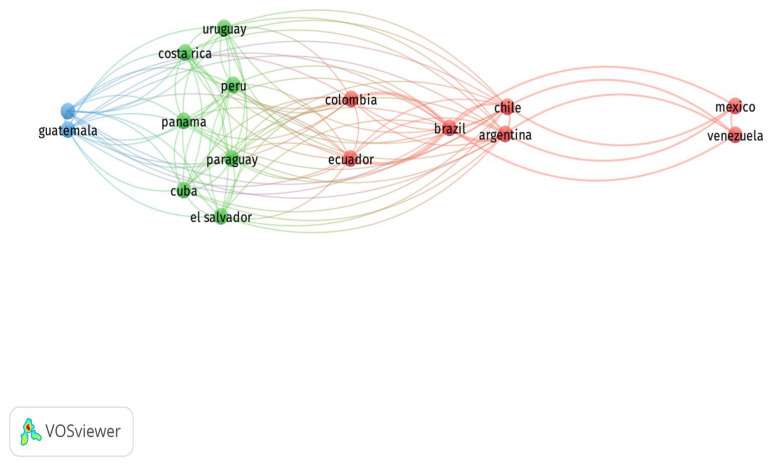
Network of collaborations between countries in the field of respiratory vaccines in the Web of Science database.

**Figure 2 vaccines-13-00240-f002:**
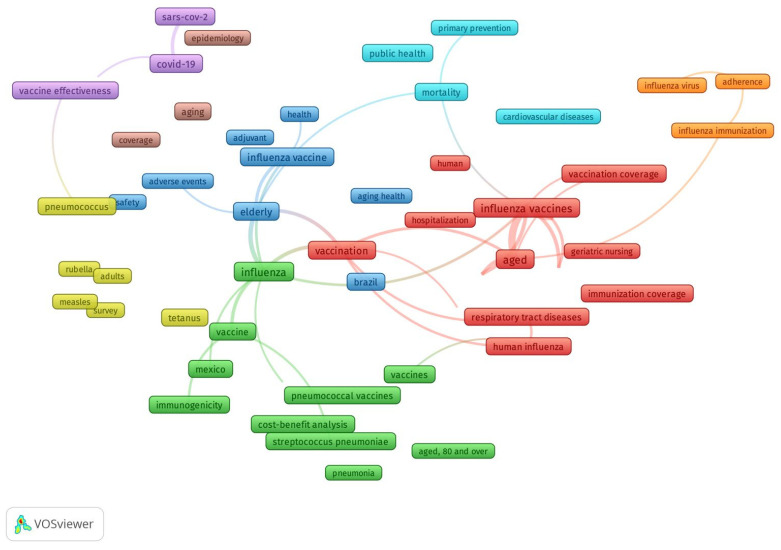
Keyword co-occurrence analysis in the field of respiratory vaccines in Web of Science.

**Figure 3 vaccines-13-00240-f003:**
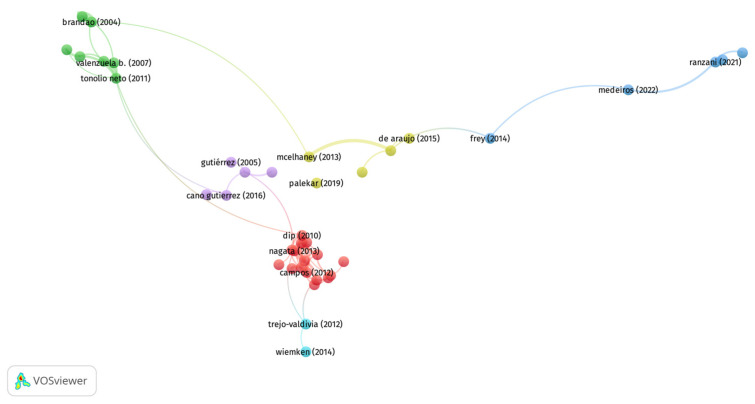
Bibliographic coupling analysis in the field of respiratory vaccines in Web of Science [29,30,32,34,35,39,40,41,42,43,44,45,46,47,48,49,50,51,52].

**Table 1 vaccines-13-00240-t001:** Most impactful journals in the field of respiratory vaccines in the Web of Science.

Rank	Journal Title	H-Index	TC	NP	Citations per Publication	Scope
1	*Vaccine*	7	260	10	26	Vaccines and infectious diseases
2	*Cadernos de Saude Publica*	5	128	13	9.9	Public health
3	*Ciencia and saúde coletiva*	5	36	7	5.1	Public health
4	*Epidemiologia e servicos de saúde*	4	42	6	7	Public health
5	*Plos one*	4	111	6	18.5	Public health
6	*Human vaccines and immunotherapeutics*	3	25	3	8.3	Vaccines and infectious diseases
7	*Revista de saude publica*	3	84	3	28	Public health
8	*Revista panamericana de salud publica-pan american journal of public health*	3	45	3	15	Public health
9	*Salud publica de mexico*	3	34	5	6.8	Public health
10	*Brazilian journal of infectious diseases*	2	17	2	8.5	Vaccines and infectious diseases

**Table 2 vaccines-13-00240-t002:** Most impactful journals in the field of respiratory vaccines in the Web of Science, by country.

Rank	Country	TC	NP	NPc	S-NPc	M-NPc	MCP_Ratio	Total Link Strength
1	Brazil	1081	70	63	59	4	6.3%	5
2	Mexico	434	14	9	8	1	11%	1
3	Colombia	208	6	2	1	1	50%	2
4	Argentina	105	11	3	3	0	0%	3
5	Chile	56	5	2	2	0	0%	2
6	Ecuador	40	2	1	0	1	100%	2
7	Guatemala	32	2	1	0	1	100%	2
8	Honduras	32	2	0	0	0	0%	2
9	Costa Rica	30	1	0	0	0	0%	1
10	Cuba	30	1	0	0	0	0%	1

The following countries outside Latin America and the Caribbean were removed for this table: the USA, Switzerland, Belgium, Italy, and Spain. NPc—number of publications with the corresponding authors; S-NPc—number of publications with the corresponding authors without international collaboration; M-NPc—number of publications with the corresponding authors with international collaboration.

**Table 3 vaccines-13-00240-t003:** Most impactful journals in the field of respiratory vaccines in the Web of Science.

Rank	Title	Author, Year (Country)	Global Citations	Total Citations	GC/TC Ratio	Vaccine—Theme (Design)
Based on Global Citations						
1	Social determinants of health and seasonal influenza vaccination in adults ≥65 years: a systematic review of qualitative and quantitative data	Nagata, 2013 (International) [29]	211	7	3.3%	Influenza—Public Health (Evidence synthesis)
2	Effectiveness of the CoronaVac vaccine in older adults during a gamma variant associated epidemic of COVID-19 in Brazil: test negative case-control study	Ranzani, 2021 (Brazil) [30]	189	4	2.1%	COVID-19—Effectiveness (Observational)
3	Remodeling of the Immune Response With Aging: Immunosenescence and Its Potential Impact on COVID-19 Immune Response	Leite Cunha, 2020 (International) [31]	149	0	0%	COVID-19—Immunology (Review)
4	AS03-adjuvanted versus non-adjuvanted inactivated trivalent influenza vaccine against seasonal influenza in elderly people: a phase 3 randomized trial	McElhaney, 2013 (International) [32]	121	4	3.3%	Influenza—Effectiveness (Experimental)
5	Comparison of the safety and immunogenicity of an MF59^®^-adjuvanted with a non-adjuvanted seasonal influenza vaccine in elderly subjects	Frey, 2014 (International) [33]	116	2	1.7%	Influenza—Effectiveness (Experimental)
Based on local citations						
1	Vacinação contra influenza em idosos: prevalência, fatores associados e motivos da não-adesão em Campinas, São Paulo, Brasil	Francisco, 2011 (Brazil) [34]	29	17	58.6%	Influenza—Public Health (Observational)
2	Fatores associados à vacinação contra influenza em idosos em município do Sudeste do Brasil	Donalisio, 2006 (Brazil) [35]	31	16	51.6%	Influenza—Public Health (Observational)
3	Impacto da vacinação contra influenza na mortalidade por doenças respiratórias em idosos	Francisco, 2005 (Brazil) [36]	27	15	55.6%	Influenza—Effectiveness (Observational)
4	Factors associated with vaccination against influenza in the elderly	Francisco, 2006 (Brazil) [37]	31	14	45.2%	Influenza—Public Health (Observational)
5	Factors associated with influenza vaccination among elderly in a metropolitan area in Southeastern Brazil	Lima-Costa, 2008 (Brazil) [38]	26	14	53.9%	Influenza—Public Health (Observational)

## Data Availability

Dataset available on request from the authors.

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
