# Peer review of "Respiratory Vaccines in Older Adults: A Bibliometric Analysis and Future Research Agenda"

_vaccines, 2025, doi:10.3390/vaccines13030240_

Round 1

Reviewer 1 Report

Comments and Suggestions for Authors

General Comments

This paper investigates and offers a thorough bibliometric analysis of respiratory vaccines for older adults in Latin America and the Caribbean (LAC). The relevant and timely study addresses critical public health challenges, particularly as the region grapples with vaccine disparities and emerging diseases. However, several areas need further clarity and expansion to enhance the paper's rigor and relevance.

Major Comments

  1. While identifying emerging themes like older adults in Latin America and the Caribbean (LAC), a deeper exploration of how these themes align with global trends in geriatric vaccination research would strengthen the paper's relevance.
  2. What are the contributions to the literature? More theoretical and/or empirical support is needed to strengthen the contributions and implications of the research results. The authors should address more papers in this paper.
  3. The study is limited to Latin America and the Caribbean (LAC). While this focus is valuable for localized insights, discussing how findings might generalize to other regions with differing demographics or healthcare infrastructures is recommended. The authors can describe the vaccination rates over the years compared with other areas. The paper highlights Brazil's dominance in research output. To add depth, discuss barriers other LAC countries face and strategies to mitigate these challenges. For example, how might funding disparities or lack of research infrastructure contribute to the observed gaps?
  4. The methodology section would benefit from greater detail, particularly about data cleaning and the decision to exclude certain databases. This would bolster the study's reproducibility and credibility. Expanding on how this gap could be addressed in future interventions would add value.
  5. Separating results from implications for practice and future research would improve readability and clarity. For instance, actionable steps for healthcare managers or policymakers could be highlighted distinctly.
  6. Given the unique demographic and healthcare disparities in LAC, there is an opportunity to adapt existing theories or develop new frameworks tailored to the region’s context. For instance, integrating socio-economic and cultural factors unique to LAC into vaccination behavior models could yield more region-specific insights

Minor Comments

  1. The presentation results of journal rankings and collaboration networks could be reformatted to emphasize key insights, such as trends in publication impact or the strength of specific partnerships.
  2. In addition, the expression method of the table can also be considered to be displayed from different role perspectives so that subsequent research can continue similar discussions.

Author Response

Dear Editors,

Vaccines Editorial Office

Manuscript ID: vaccines-3420924

Thank you for the opportunity to address your comments regarding our manuscript.

We appreciate your thorough review and valuable feedback, which has helped us improve the

quality and clarity of our work. We have carefully considered each point raised and have made

the following revisions to address your concerns:

  • While identifying emerging themes like older adults in Latin America and the Caribbean (LAC), a deeper exploration of how these themes align with global trends in geriatric vaccination research would strengthen the paper's relevance. What are the contributions to the literature? More theoretical and/or empirical support is needed to strengthen the contributions and implications of the research results. The authors should address more papers in this paper.

Response: We appreciate the comment. We have added additional information regarding this in the introduction and the discussion.

  • The study is limited to Latin America and the Caribbean (LAC). While this focus is valuable for localized insights, discussing how findings might generalize to other regions with differing demographics or healthcare infrastructures is recommended. The authors can describe the vaccination rates over the years compared with other areas. The paper highlights Brazil's dominance in research output. To add depth, discuss barriers other LAC countries face and strategies to mitigate these challenges. For example, how might funding disparities or lack of research infrastructure contribute to the observed gaps?

We appreciate this observation. However, we have not found precise information on vaccination rates by country or region according to life stages or specifically in older adults. It is reported that in Latin America, these rates are lower than in North America and Europe. This topic has been addressed in the discussion.

  • The methodology section would benefit from greater detail, particularly about data cleaning and the decision to exclude certain databases. This would bolster the study's reproducibility and credibility. Expanding on how this gap could be addressed in future interventions would add value.

Response: Thank you for this observation. While it is important to search multiple databases for other types of literature reviews, this would be detrimental to bibliometrics. Hence, we decided to use only Web of Science, as described in the methodology. We hope that further methods may be developed to merge databases in the future.

  • Separating results from implications for practice and future research would improve readability and clarity. For instance, actionable steps for healthcare managers or policymakers could be highlighted distinctly.

Response: We appreciate this observation. We have added additional information regarding this in the discussion section.

  • Given the unique demographic and healthcare disparities in LAC, there is an opportunity to adapt existing theories or develop new frameworks tailored to the region’s context. For instance, integrating socio-economic and cultural factors unique to LAC into vaccination behavior models could yield more region-specific insights.

Response: We appreciate this comment. We have added additional information regarding this in the discussion section.

  • The presentation results of journal rankings and collaboration networks could be reformatted to emphasize key insights, such as trends in publication impact or the strength of specific partnerships. In addition, the table’s expression method can be considered to display different role perspectives so that subsequent research can continue similar discussions.

Response: While we acknowledged this recommendation, we consider it unfeasible due to the classic formatting of bibliometrics.  

Best regards

Javier A. Flores-Cohaila

Reviewer 2 Report

Comments and Suggestions for Authors

The authors propose a review of the literature regarding vaccines in Latin America.

1.     The study appears to have been carried out by 9 authors, but not all of them appear to have played an essential role in the review. For example, L.M.G.R. appears to have managed the acquisition of funds, but the research was not funded. I advise the authors to review the names of the authors, indicating those who actually worked.

2.     In the Introduction, the authors outline the importance of respiratory infections in older adults worldwide but say nothing about the impact of these infections on older adults in Latin America.

3.     In the Introduction, the authors list numerous resources for controlling respiratory infections in the elderly and put vaccines last. There is a lack of bibliographical indications on the usefulness of vaccines in the elderly

4.     The shift from the global to the local situation is sudden and unmotivated. The authors inform us only of the fact that in Latin America there are deficiencies in training of health personnel and disparities in health services. It is not clear why these facts should motivate the research presented here. The fact that no one has done this study is not enough.

5.     One unclear aspect of the research is whether there are studies conducted in collaboration by researchers from Latin America and other countries. It seems strange that there is no collaboration, but if so, the authors should emphasize this point.

6.     Similarly, it seems strange that there is a complete lack of studies on genomics and new vaccines. Perhaps Latin America is completely lacking in research institutes on these topics? Or were existing research not included in the search because the search string excluded it? The authors should clarify this point.

Author Response

(The authors gave the same response as above.)

Round 2

Reviewer 1 Report

Comments and Suggestions for Authors

The author has made the necessary corrections and should meet the publication requirements.

Reviewer 2 Report

Comments and Suggestions for Authors

the authors have revised the manuscript